# A message-passing multi-task architecture for the implicit event and polarity detection

**Chunli Xiang**[1], **Junchi Zhang**[2], **Donghong Ji**[1]*

**1** Key Laboratory of Aerospace Information Security and Trusted Computing, Ministry of Education, School of Cyber Science and Engineering, Wuhan University, Wuhan, Hubei, China, **2** School of Computer Science and Engineering, Wuhan Institute of Technology, Wuhan, Hubei, China

* dhji@whu.edu.cn

**Data Availability Statement:** The CLIPEval dataset can be downloaded at: https://alt.qcri.org/semeval2015/task9/.

**Funding:** The author(s) received no specific funding for this work.

## Abstract

Implicit sentiment analysis is a challenging task because the sentiment of a text is expressed in a connotative manner. To tackle this problem, we propose to use textual events as a knowledge source to enrich network representations. To consider task interactions, we present a novel lightweight joint learning paradigm that can pass task-related messages between tasks during training iterations. This is distinct from previous methods that involve multi-task learning by simple parameter sharing. Besides, a human-annotated corpus with implicit sentiment labels and event labels is scarce, which hinders practical applications of deep neural models. Therefore, we further investigate a back-translation approach to expand training instances. Experiment results on a public benchmark demonstrate the effectiveness of both the proposed multi-task architecture and data augmentation strategy.

## Introduction

Sentiment analysis is a hot research field in Natural Language Processing (NLP). It has been advanced by the recent progress of machine learning and deep learning techniques. While the bulk of current research has been done on explicit sentiment analysis [1–7], less focus has been paid to detect implicit sentiment in the literature. There are two examples showing the differences between explicit and implicit sentiment polarities.

s1: *I ate a bad McRib this week*

s2: *In within a month, a valley formed in the middle of the mattress.*

Compared with sentiments with explicit subjective expressions (e.g., 'bad' in s1), implicit sentiments mean no clear sentimental terms contained in reviews, requiring the polarity label inferred from sentence contexts. Consider s2, a negative polarity towards the mattress is implied by metaphorizing it's quality to a valley. Previous methods for this problem rely heavily on hand-crafted features [8–10]. Inspired by dictionary-based sentiment analysis, Feng et al. (2013) analogously create a connotation lexicon where words are objective on the surface but connotative to positive or negative sentiments [8]. Schouten et al. (2014), on the other hand, utilize labeled implicit features and notional words to build a co-occurrence matrix. The

**Competing interests:** The authors have declared that no competing interests exist.

features that gain the highest co-occurrence score of the notional words are used as the implicit features of the sentence [9]. However, the above attempts focus on sentence-level overall sentiments without taking fine-grained polarities into account. Latter, there have been growing interests considering the fine-grained implicit sentiments recognition, where thorough analyses are introduced in addition to polarity classifications [11–13]. Chen et al. (2016) construct an implicit opinion corpus annotated with aspect and polarity labels [11]. They detected implicit opinions with surrounded opinion terms. In the line of their work, Liao et al. (2019) focus on fact-implied implicit sentiment sentences [13]. They proposed a multi-way Convolutional Neural Network to fuse text n-gram features, target features, and background features to identify implicit polarities. In the present paper, we turn our attention to joint learning of text sentiments and corresponding events. We take an example to indicate the benefits of simultaneously learning these two tasks. Consider the sentence 'I hiked mountains in Colorado with them', the event is 'I hiked mountains' that describes the objective, factual statements, which is the reporter's outdoor activity experience. The sentiment orientation towards the event is positive since the need to have outdoor activities is probably satisfied. In essence, our task is generally divided into two subtasks: (1) event detection. We treat it as a multi-class classification problem that clusters the expression into generic 8 types of events(described in detail later). (2) the identification of implicit sentiment related to the event. It involves associating each sentence with a class of the polarity value (POSITIVE, NEGATIVE, or NEUTRAL).

Event identification and sentiment detection can be treated as two independent tasks(as illustrated in Fig 1(a)). In this case, the model does not make full use of correlated information among subtasks. Such pipeline methods (shown in Fig 1(b)) [14], which identify the event first, followed by detecting polarities toward the event, suffering from the limitation that if correct monitoring is not implemented regarding the event would lead to incorrect predictions of polarities, resulting in performance degradation. Since these two subtasks are interrelated, we consider the joint learning approach that uses a single network to model two subtasks in parallel, which are referred to as multi-task learning (MTL)(shown in Fig 1(c)). MTL has been

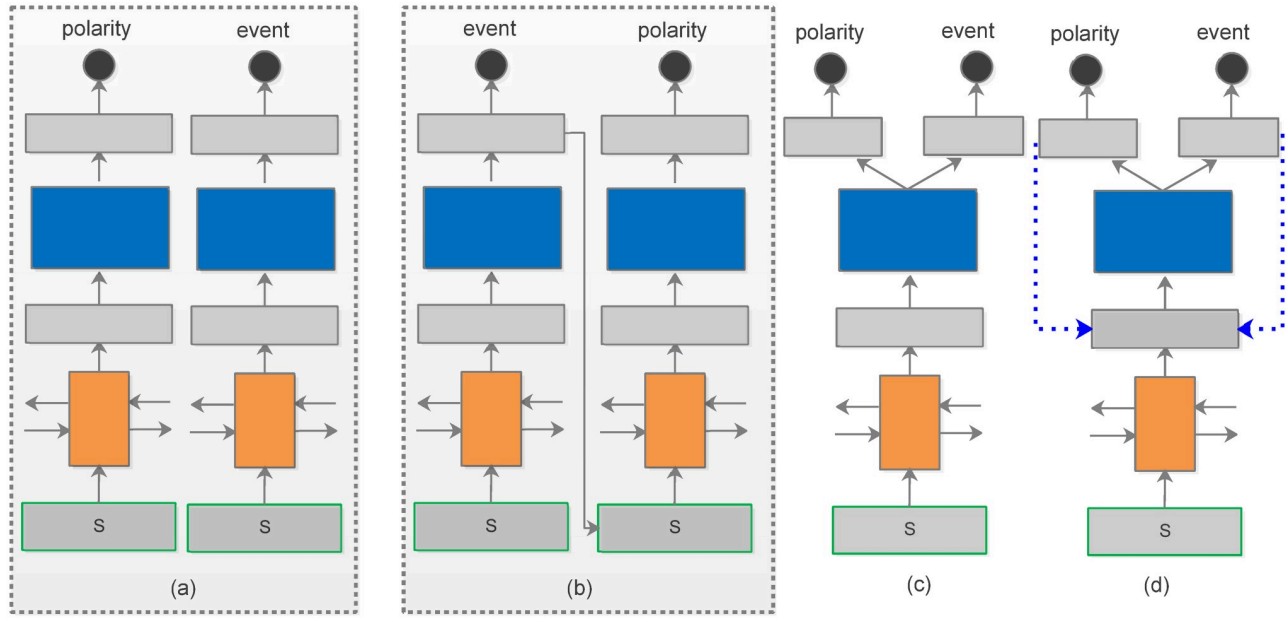

**Fig 1. A set of proposed approaches for event and polarity identification, where s represent the input sequence.**

demonstrated to present the ability to give remarkable predictive performance when involves in several highly relevant tasks [15–18]. Specifically, MTL contains a shared component and several private output layers to model multi-source inputs. Different losses are combined with weight to regulate shared and task-specific parameters in MTL. In practice, traditional approaches usually simulate task relevance by defining task common neural structures, and the parameters are then updated by back-propagation. To improve the performance of MTL, some researchers introduce external knowledge such as commonsense databases, which enable the model to exploit real-world information. Such knowledge is designed manually and has a strong dependence on domain experts, which is labor-intensive and time-consuming.

Different from previous approaches, we present an iterative learning framework based on MTL via a message-passing mechanism(as shown in Fig 1(d)). We refer it to message-passing multi-task learning(MPMTL), which sends useful information generated in the training process back to the model, allowing different subtasks to interact with each other. Specifically, we treat the task-specific output vectors generated from the previous iteration as knowledge and fuse it into the next step representation. This knowledge is updated and changed in the process of training, allowing interactions between subtasks. By involving a message-passing mechanism into the MTL, the proposed model not only adds supervision for learning more robust deep representations but also can integrate information across domains.

There has been a public dataset (named CLIPEval dataset) for implicit polarity and event detections [19]. However, the dataset is limited in size, which tends to cause the overfitting issue in deep neural network training. In computer vision, data augmentation technologies are widely applied to generate auxiliary training examples [20–22]. In NLP, back-translation has been proven to be effective in augmenting diverse instances [23–25]. We borrow this idea and translate training sentences into pivot languages. Then sentences in targeted languages are translated back to English, which attempts to rephrase a sentence while holding its original semantics. Keep the original sentence and the back-translated sentence belong to the same class, which can effectively double the size of the original dataset, hoping that it brings the linguistic diversity of realistic variables and reduces dependence on data collection. The experiments illustrate that the data augmentation strategy can significantly improve the generalization of our model.

In summary, our contributions are threefold:

- We propose a multi-task learning framework to identify events and corresponding implicit polarity simultaneously.

- The key contribution of this work is to exploit a new message-passing architecture to fuse dark knowledge from previous iterations into the model to achieve an optimal result iteratively.

- To expand the samples' size and diversity in the training process, we use a back-translation technology to expand the training set.

## Related work

### Event-based implicit polarity analysis

There has been a large body of research in sentiment analysis over the last decade. A bulk of current research was devoted to recognizing explicit sentiment containing opinion words [1, 3, 4, 26, 27]. However, many factual text types often contain implicit expressions of sentiment as well, i.e., objective, factual statements that are connotative to positive or negative sentiments.

Although some studies have explored the formalization and statistical representations of these implicit sentiments [11, 13, 28, 29], it is still a great challenge to identify them automatically. Kauter et al. (2015) show a new topic-specific implicit and explicit sentiment analysis in financial newswire texts [30]. The overall sentiment expressed by a sentence towards the given company was determined by aggregating the polarity scores of all relevant polar expressions in that sentence. Russo et al. (2015) propose a public dataset (named CLIPEval) for identifying the implicit polarity of events [19]. They illustrate that many attitudes are conveyed implicitly, and the expression event is vital for inferring the implicit sentiment. Dragoni et al. (2015) exploit Information Retrieval (IR) techniques for representing information about the linguistic structure of sentences and implements a variation of classic IR scoring formula to infer the event and the implicit polarity [2]. In the present paper, we formalize event recognition and sentiment recognition into two classification tasks and use a multi-task learning framework to make two subtasks better interact with each other.

## Multi-task learning

Multi-task learning(MTL) has attracted much attention in recent years. It makes full use of the interactive information of multiple tasks to improve both tasks simultaneously [31]. The applications of the MTL have been proven to be effective in many NLP problems, such as sequence labeling [32], text classification [16, 33], and text summarization [34], etc. Liu et al. (2016) propose three different information-sharing mechanisms based on LSTM to model the text [33], which is the first to utilized LSTM to construct a multi-task learning framework for text classification. Zheng et al. (2018) employ attention mechanisms to select the task-specific information from the shared representation layer [16]. These multi-task learning schemes typically use a generic shared component to model the text representation and several private learning components to learn the task-specific representations for task predictions. In our work, we use a message-passing mechanism based on the MTL framework, which exploits useful information generated in the training process to update the context representation.

## Message-passing mechanism

The message-passing mechanism has been studied in computer software [35] and NLP [36]. Recently, Gilmer et al. (2017) apply neural message-passing algorithms allowing long-range interactions between nodes in the graph [37]. More recently, He et al. (2019) introduce a message-passing architecture to pass the information to different subtasks through a shared set of latent variables [38], which enable the correlation of different subtasks to be better utilized. Our design of the message-passing mechanism was inspired by [38]. The difference is that we take task-specific representations into account as prior knowledge during the training process rather than the label probability distribution. Furthermore, we use the back-translation strategy to obtain additional training samples.

## Model

Given a sequence of tokens $(x_1, x_2, \ldots, x_i, \ldots, x_n)$, where $x_t$ is the $t - th$ token in the sequence, our task aims to identifying the event label and the corresponding implicit polarity label. We formulate the problem as two multi-class classification tasks: event classification and implicit sentiment detection.

In the following, we introduce the design and implementation of our model in detail.

## Attention-based recurrent neural network for text classification

We use LSTM to learn latent word semantics [39], since LSTM is instrumental in learning long-term dependencies and can avoid the gradient vanishing and expansion problems. To capture the most important words related to the target subtask, we add an attention mechanism on the top of the LSTM [7]. The attention mechanism is adopted in various NLP tasks for their superior performance to capture the relation words no matter their distance in text [40].

Specifically, an embedding layer firstly projects the source sentence into a sequence of continuous representations $(e_1, e_2, \ldots, e_t, \ldots, e_n)$, here $e_t$ is the word vector of the $t - th$ word in the sentence projected by a word embedding layer. To gain dependencies between adjacent words in forward and backward directions, we use a bi-directional LSTM (BiLSTM) to obtain sequence hidden states $H = (h_1, h_2, \ldots, h_t, \ldots, h_n)$, here $h_t = [\overrightarrow{h_t}, \overleftarrow{h_t}]$ is the hidden representation of $e_t$, $\overrightarrow{h_t}$ and $\overleftarrow{h_t}$ are the corresponding chronological order hidden state and reverse order hidden states, respectively:

$$\overrightarrow{h_t} = \overrightarrow{LSTM}(h_{t-1}, e_t, \theta_1)$$
$$\overleftarrow{h_t} = \overleftarrow{LSTM}(h_{t-1}, e_t, \theta_2)$$

$$(1)$$

where $\theta_1$ and $\theta_2$ are the model parameters. The formulas of $\overrightarrow{LSTM}$ are formally presented as follows:

$$i_t = \sigma(W_{ii} \cdot e_t + b_{ii} + W_{hi} \cdot \vec{h}_{t-1} + b_{hi})$$

$$f_t = \sigma(W_{if} \cdot e_t + b_{if} + W_{hf} \cdot \vec{h}_{t-1} + b_{hf})$$

$$g_t = \tanh(W_{ig} \cdot e_t + b_{ig} + W_{hg} \cdot \vec{h}_{t-1} + b_{hg})$$

$$(2)$$

$$o_t = \sigma(W_{io} \cdot e_t + b_{io} + W_{ho} \cdot \vec{h}_{t-1} + b_{ho})$$

$$c_t = f_{t^\circ} c_{t-1} + i_{t^\circ} g_t$$

$$\vec{h}_t = o_{t^\circ} \tanh(c_t)$$

where $W$ and $b$ are the model parameters, $\cdot$ represents the dot multiplication, $\circ$ is the Hadamard product. $tanh$ is a non-linear function and $\sigma$ is the sigmoid function. The attention mechanism takes the entire LSTM hidden state H as input and outputs a weight vector $a$:

$$a = \text{softmax}(W_2 \cdot \tanh(W_1 \cdot H^T))$$

$$(3)$$

where $W_1$ and $W_2$ are weight matrices, and the *softmax function* ensures all the computed weights sum up to 1.

Then we use $a$ to linearly combine BiLSTM hidden states $H$ to choose the most important features for producing the next layer representation:

$$F = a \cdot H$$

$$(4)$$

Then we use a full-connected layer followed by a softmax non-linear layer to calculate the probability distribution over classes. That is,

$$\hat{y} = \text{softmax}\,(W \cdot F + \mathbf{b}) \tag{5}$$

where $W$ and $b$ are the weight matrix and bias, respectively. The loss function for the target subtask can be formulated of using cross-entropy error between the predicted distribution and the gold distribution:

$$Loss(\hat{y}, y) = -\sum_{i=1}^{N}\sum_{j=1}^{C} y_i^j \log\,(\hat{y}_i^j) \tag{6}$$

where $C$ is the label type of target subtask. $y_i$ denotes the ground-truth label, represented by a one-hot vector, $\hat{y}_i \in R^{|C|}$ is the predicted probability distribution. Since this baseline approach treats the identification of the event and the polarity as two independent tasks, their model parameters need to be trained from scratch, which is insufficient to introduce interactive knowledge between tasks.

## Multi-task learning for text classification

MTL is a useful framework to make use of the correlation information of related tasks to improve generalization performance simultaneously (shown in Fig 1(c)). This framework mainly contains two components, which are the shared structure to learn common representations and several task-specific structures to learn task-specific representations, respectively. To facilitate illustration, we denote $D$ as a dataset with $n$ instances, which can be formally written as $D = \{(x_i, y_i^p, y_i^e)\}_{i=1}^{n}$, where $x_i$ is the $i$-th input sequence and $y_i^p$ and $y_i^e$ are the corresponding sentiment and event label. To encourage information exchange among different subtasks, we use $F$ (which is defined as Eq (4))to transform the input text into the shared context representation. Based on that, we exploit a separate multi-layer perceptron decoder (containing a fully connected layer and a softmax layer) to transform the context representation into category probability for each subtask:

$$
\begin{aligned}
\hat{y}^p &= \text{softmax}\,(W_p \cdot F + \mathbf{b_p}) \\
\hat{y}^e &= \text{softmax}\,(W_e \cdot F + \mathbf{b_e})
\end{aligned}
\tag{7}
$$

where $W_p$, $W_e$, $b_p$ and $b_e$ are task-specific parameters. We expect our model to learn the shared representations that benefit all two subtasks and capture the most important and salient clues of the text, i.e., parts of the text that mention the event and corresponding polarity. The model parameters are trained to minimize the cross-entropy error of the predicted and gold distributions for all subtasks. The final loss function can be computed as:

$$Loss = \lambda_1 Loss(\hat{y}^p, y^p) + \lambda_2 Loss(\hat{y}^e, y^e) \tag{8}$$

where $\lambda_1$ and $\lambda_2$ are hyper-parameters, $Loss(\hat{y}^p, y^p)$, and $Loss(\hat{y}^e, y^e)$ are calculated using Eq (6), which signify the loss of the event classification and polarity detection, respectively.

## Message-passing multi-task learning model

Based on the multi-task learning model, a common way of information fusion is to perform a naive concatenation of distilled features from different semantic layers. In our model, via message-passing mechanism, we send the task-specific latent features generated at $t-1$-th time step as the knowledge back to the model to update the shared latent sentence representation $F_t$

at $t$-th time step. Specifically, the shared latent sentence representation $F_t$ is updated by:

$$F_t = [F_t \oplus Vp_{t-1} \oplus Ve_{t-1}] \tag{9}$$

where $\oplus$ represents concatenation operation, $F_t$ is defined as Eq (4).

We use $Vp_{t-1}$ and $Ve_{t-1}$ to represent the task-specific knowledge at $t-1$ time step:

$$\begin{aligned} Vp_{t-1} = & \quad (W_p \cdot F_{t-1} + \mathbf{b_p}) \\ Ve_{t-1} = & \quad (W_e \cdot F_{t-1} + \mathbf{b_e}) \end{aligned} \tag{10}$$

where $W_p$ and $W_p$ denote weight matrices, $b_p$ and $b_e$ denote bias terms. Using the message-passing mechanism, two kinds of representations are jointly refined iteratively. The task-specific features can guide and prompt the context representation learning, which is then used with two separate multi-layer perceptrons for different subtasks prediction.

## Message-passing multi-task learning model with data augmentation

Previous studies show that using the data augmentation approach can significantly improve the model's generalization ability and reduce the over-fitting risk. It is particularly effective when the training data is inadequate because augmented data can bring valuable rewriting information that is not embraced by the original data. Another advantage of the data augmentation approach is that it does not modify the existing model architecture. However, previous studies tend to train the model with original training data and augmented data separately. In the present work, our message-passing mechanism opens up the possibility of making the interaction between the original data and the augmented data in the training process, leading to the model being more tolerant of the noise in the augmented data.

Here we describe how to use the data augmentation approach in MPMTL. First, we exploit machine translation tools to perform back translation to generate the augmented data. Since augmented data usually contains noise and is less valuable than original training data. During training, we give the original sentences and their augmented data different weight in one mini-batch. In particular, we set a larger ratio of $p$ for choosing the original sentences and define the augmented policy as a list of $m$ pivot languages. In the back translation process, we translate the original sentences to a target language and then translate them back to English. For each step $t$, we not only feed the input sentence to the MPMTL, but also take previous $Vp_{t-1}$ and $Ve_{t-1}$ (defined as Eq (10) as task-related information to update the context representation $F_t$ (defined as Eq (4), which then used as shared features for the current step predictions.

## Experimental settings

### Data and data augmentation settings

To demonstrate the effectiveness of the proposed approach, we evaluate the implicit polarity and event identification task on a publicly available dataset: CLIPEval corpus, which is freely available at http://alt.qcri.org/semeval2015/task9/. The dataset contains self-reported reviews collected from English Gigaword corpus [41]. Each sample is annotated with an event class label and a polarity label concerning the event. Table 1 shows the statistical distribution of the dataset, containing 1, 280 instances for training and 371 for testing. To further improve the performance of the MPMTL model, we apply a back-translation technology to increase the scale of the training dataset. We exploit machine translation tools to perform back translation. Specifically, We first use Google Translation to translate the original sentence to the pivot language and then adopt Baidu Translation to translate it back into English. Spanish, Dutch, and

**Table 1. Statistical information of dataset.**

| Dataset | Training set | | | | Test set | | | |
|---|---|---|---|---|---|---|---|---|
| Event Class | Positive | Negative | Neutral | Total | Positive | Negative | Neutral | Total |
| (FEAR_OF)_PHYSICAL_PAIN | 19 | 131 | 10 | 160 | 10 | 30 | 5 | 45 |
| ATTENDING_EVENT | 83 | 35 | 42 | 160 | 29 | 5 | 11 | 45 |
| COMMUNICATION_ISSUE | 21 | 120 | 19 | 160 | 8 | 29 | 7 | 44 |
| GOING_TO_PLACES | 55 | 72 | 33 | 160 | 22 | 23 | 3 | 48 |
| LEGAL_ISSUE | 24 | 115 | 21 | 160 | 5 | 27 | 3 | 45 |
| MONEY_ISSUE | 20 | 109 | 31 | 160 | 12 | 27 | 12 | 51 |
| OUTDOOR_ACTIVITIES | 125 | 18 | 17 | 160 | 34 | 4 | 8 | 46 |
| (PERSONAL_CARE | 88 | 40 | 32 | 160 | 24 | 10 | 13 | 43 |

German are chosen as pivot languages. Importantly, the back-translation strategy is only used during training, not for evaluation.

## Hyperparameters

We use 90% of the training data for training and the remaining 10% for validation. We evaluate the performance of the model on the validation set to obtain the final model hyper-parameters. We use Bert embedding in flair library, which is publicly available at https://github.com/flairNLP/flair as the word embeddings for all models. We don't fine-tune it during model training. Word embedding size is set to 748. The mini-batch size is set to 16. The dimension of the hidden state of the BiLSTM layer is set to 100. The fully connected layer to calculate class scores is set to 100 hidden units with a dropout rate of 0.6 [42]. The hyperparameters $\lambda_1$ and $\lambda_2$ of loss function are both set to 0.5. Models are trained via Adam optimizer with the learning rate of 1e-3 [43]. For our proposed MPMTL model, we apply a message-passing mechanism to fuse a schedule of knowledge into the model for predicting different subtasks. This schedule of knowledge generates iteratively in the training process. It is reasonable that the performance on the test set will change as the iteration runs. We thus set the iteration number to four in the test stage as it reaches the best performance. We set the probability of choosing the original sentence to 0.6 when applying the data augmentation strategy in the training process.

## Evaluation metrics

Following Russo et al. (2015), we evaluate the experimental results in terms of micro-average precision, recall, and F1 score [19], where A-e(A-p), R-e(R-p), and F1-e(F1-p) denotes precision, recall, and F1 score, respectively. The overall performance by jointly considering two subtasks about micro-average precision, recall, and F1 score are denoted as A-o, R-o, and F1-o, respectively.

## Baseline models

To show the effectiveness of our proposed model, we chose three state-of-the-art models as baselines.

- ATTLSTM: For each subtask, it contains an embedding layer, A BiLSTM layer, and an attention mechanism based on the BiLSTM layer to model the sentence representation and followed by a multi-layer perceptron(containing a full-connected layer and a softmax layer) for subtask predictions [7].

- PIPELINE: performing the event detection and polarity identification tasks in a pipeline manner that detects the event first and then combines the sentence and the event as the input to identify the corresponding sentiment [14].

- MTL: using a front-end attention-based BiLSTM structure as the shared structure to learn the sentence's embedding, then performing two multi-layer perceptrons to identify events and sentiments [16].

## Results and analysis

### Comparison with baselines

In this section, we analyze and discuss the effects of some of the design choices in setting up MPMTL and probe some aspects to answer why the method is effective. We compare the results of our MPMTL model with the baselines in Table 2. Note that ATTLSTM treats the identification of the event and the polarity as two independent tasks. And the event identified by the ATTLSTM is used as an extra source combining with the sentence to identify polarity for PIPELINE. The notation _ refers to the results that were not given in the original paper. The results evaluate both the event and the polarity. SHELLFBK system only submitted the polarity and overall results for SemEval 2015 task 9. The basic attention BiLSTM(ATTLSTM) model achieves an F1-o score of 72.72%. PIPELINE model shows that the polarity performance can achieve an f1-o score of 73.69% if the gold event information is provided. Compared with the PIPELINE model, multi-task learning improves performance by 0.79% in F1-o, which shows that multi-task learning is effective for fine-grained implicit sentiment analysis verifying the advantage of our proposed framework. As we expected, our MPMTL model outperforms the baselines and achieves an F1-o score of 75.02%. MPMTL utilizes knowledge generated from different tasks. This shows that domain-specific knowledge is beneficial, and both joint training and domain-specific features effectively transfer such knowledge. Furthermore, the best generalization gains 75.95% of F1-o when applying the data augmentation strategy on MPMTL.

### Attention distribution exploration

To understand how the proposed MTMTL work compared with the vanilla MTL model, we visualize the attention weights of each word in Fig 2. Here, we randomly sample the test set and analyze the attention weight changes with different approaches. We can see both models capture the informative pattern 'having lunch'. However, MTL makes a wrong prediction of the polarity. By contrast, our MPMTL makes a correct prediction. The reason may be our model fuses a schedule of task-specific knowledge that helps infer the implicit sentiment via capturing the phrase 'guardian angels'. This phrase receives more attention weights, which are

**Table 2. Experimental results based on CLIREval dataset.**

| Models | P-e(%) | R-e(%) | F1-e(%) | P-p(%) | R-p(%) | F1-p(%) | P-o(%) | R-o(%) | F1-o(%) |
|---|---|---|---|---|---|---|---|---|---|
| SHELLFBK | _ | _ | _ | 56.00 | 56.00 | 54.00 | 36.00 | 27.00 | 29.00 |
| ATTLSTM | 88.53 | 87.60 | 87.65 | 82.43 | 82.48 | 82.21 | 75.06 | 72.51 | 72.72 |
| PIPELINE | 88.53 | 87.60 | 87.65 | 82.81 | 82.75 | 82.68 | 75.47 | 73.85 | 73.69 |
| MTL | 89.35 | 89.22 | 89.18 | 82.94 | 83.02 | 83.17 | 76.79 | 74.39 | 74.48 |
| MPMTL | 90.06 | 89.95 | 89.10 | 83.01 | 83.29 | 83.31 | 78.57 | 74.93 | 75.02 |
| MPMTL(Aug) | 91.65 | 90.11 | 91.03 | 84.61 | 84.64 | 84.52 | 78.99 | 76.28 | 75.95 |

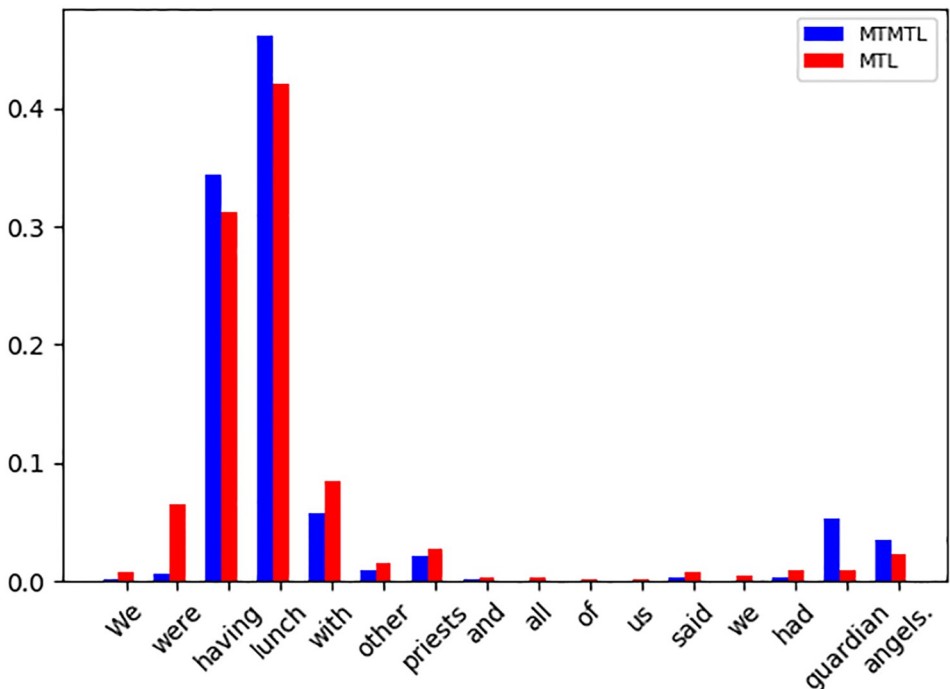

**Fig 2. The different attention weights for MTL and MPMTL models.** The X-axis represents the text tokens in chronological order. while Y-axis represents the attention weights.

important clues for predicting the polarity of the event. The above visualization shows the effectiveness of our proposed MPMTL model.

## Case study

To verify the advantage of the proposed framework, we provide a qualitative analysis to compare our MPMTL approach with ATTLSTM and MTL on the CLIREval test set. Table 3 shows three instances, along with their predicted polarities for corresponding events. $\sqrt{}$ and $\times$ denote the incorrect and correct label, respectively. For the first instance, all models give the correct 'NEGATIVE' for the event 'COMMUNICATION_ISSUE'. This is likely, due to that the words 'dispute' and 'issue' provide strong clues for event and polarity predictions. For the second instance, both ATTLSTM and MTL can still give the correct prediction of event type 'PERSO-NAL_CARE while it is difficult to infer the sentiment related to the event. The possible reason is that 'go on a diet' gives the obvious signal for predicting the correct event type. But this sentence also involves complicated semantics that both ATTLSTM and MTL models can't remove the noise information 'overweight' and 'diabetes' effectively, which leads to a false polarity

**Table 3. Case study on ATTLSTM, MTL and MPMTL approaches.**

| Approaches | ATTLSTM | | MTL | | MPMTL | |
|---|---|---|---|---|---|---|
| Examples | Event | Polarity | Event | Polarity | Event | Polarity |
| I had a dispute with Prime Minister Erbakan on this issue. | $\sqrt{}$ | $\sqrt{}$ | $\sqrt{}$ | $\sqrt{}$ | $\sqrt{}$ | $\sqrt{}$ |
| I decided to go on a diet because my overweight uncle has gotten diabetes. | $\sqrt{}$ | $\times$ | $\sqrt{}$ | $\times$ | $\sqrt{}$ | $\sqrt{}$ |
| I walked through a mountain-view unit the next day and I found the same orange-and-white color scheme, the same media console and a gener-ous balcony, but the bathroom was smaller and there was no walk-in closet. | $\times$ | $\times$ | $\sqrt{}$ | $\times$ | $\sqrt{}$ | $\sqrt{}$ |

**Table 4. Improvements with different augmentation strategies.**

| Approaches | F1-o(%) |
|---|---|
| CLIPEval dataset | 75.02 |
| +Spanish-based | 75.37 |
| +Dutch-based | 75.95 |
| +German-based | 74.62 |

prediction. For the third instance, ATTLSTM and MTL fail to predict the correct polarity, and ATTLSTM can't even predict events correctly because it pays more attention to 'but', 'bathroom', 'closet' and 'smaller', which leads to PERSONALCARE event and NEGATIVE polarity. And MTL enhances the ATTLSTM with two task-dependent components and can predict events correctly. However, there is a limitation that irrelevant words bring the noise to attention scores, such as large weights are put to 'but' and 'nothing', which adversely leads false to the polarity. In contrast, our MPMTL model successfully predicts both event and polarity for all three instances. With the help of the proposed attention mechanism and multi-task learning framework, our model can capture important clues, such as 'dispute', 'issue', 'go', 'diet', 'walked', 'mountain-view'. Furthermore, with the assistance of the message-passing mechanism, our proposed model combination considers context features learned from current iteration and task-specific latent features learned from previous iterations in an iterative interaction manner, which avoids capturing noise information, especially when texts are too long. This result shows that the self-supervised message-passing mechanism is necessary for injecting task-specific knowledge.

## The impact of data augmentation strategy

We present three pivot languages(i.e., Spanish, Dutch, and German) to conduct an exploratory study of back-translation strategies applied on MPMTL. We break the augmentation dataset into three parts: (i) Spanish-based augmented data; (ii)Dutch-based augmented data; (iii)German-based augmented data. The performances of the MPMTL model trained on different subsets of augmentation dataset combining with the original training set shown in Table 4. Adding Spanish-based augmented data only boosts performance by +0.35% F1-o comparing with the original training set. Combining with Dutch-based and Spanish-based augmented data increases the boost in performance by +0.58% F1-o. This shows exploiting Spanish and Dutch as pivot languages indeed to add quantity and variety to our training data distribution. However, applying the overall three types of augmented data decreases the performance. The main reason maybe is that the massive augmented data contain too much noise brought by the imperfect translation tool. This makes the model transfer unnecessary and even fallacious knowledge that overwhelms the original data.

## Conclusion

In this paper, we propose a MPMTL model for identifying implicit sentiments and event types for a given sentence. Based on the multi-task learning framework, our proposed MPMTL fuses the task-specific knowledge generated from the training process into the model to encourage the model to fully interact with different subtasks. Experimental results on the CLIPEval dataset demonstrate the effectiveness of our proposed model. Furthermore, data augmentation is utilized to generate samples as supplementary to the original training data, which can effectively double the size of the original dataset and achieve promising results. However, Some noisy and biased examples are inevitably introduced by the back-translation protocol due to

the imperfect translate tool performance. We believe an optimized data augmentation strategy could further improve the results and leave it to future work.

## Supporting information

**S1 Dataset. Examples and URLs of the datasets.**
(PDF)

## Acknowledgments

We would like to thank the handling editor and anonymous reviewers for their valuable and insightful comments.

## Author Contributions

**Conceptualization:** Chunli Xiang.

**Data curation:** Chunli Xiang.

**Formal analysis:** Chunli Xiang.

**Funding acquisition:** Donghong Ji.

**Investigation:** Chunli Xiang.

**Methodology:** Chunli Xiang.

**Resources:** Chunli Xiang.

**Software:** Chunli Xiang, Junchi Zhang.

**Supervision:** Donghong Ji.

**Visualization:** Chunli Xiang.

**Writing – original draft:** Chunli Xiang, Junchi Zhang.

**Writing – review & editing:** Junchi Zhang, Donghong Ji.

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
